# Structural Characterization of the Metalized Radical Cations of Adenosine ([Ade+Li-H]^•+^ and [Ade+Na-H]^•+^) by Infrared Multiphoton Dissociation Spectroscopy and Theoretical Studies

**DOI:** 10.3390/ijms242015385

**Published:** 2023-10-20

**Authors:** Min Kou, Luyang Jiao, Shiyin Xu, Mengying Du, Yameng Hou, Xianglei Kong

**Affiliations:** 1State Key Laboratory of Elemento-Organic Chemistry, Frontiers Science Center for New Organic Matter, College of Chemistry, Nankai University, Tianjin 300071, China; 2Tianjin Key Laboratory of Biosensing and Molecular Recognition, College of Chemistry, Nankai University, Tianjin 300071, China

**Keywords:** adenosine, radical, UVPD, IRMPD spectroscopy, lithiated, sodiated

## Abstract

Nucleoside radicals are key intermediates in the process of DNA damage, and alkali metal ions are a common group of ions in living organisms. However, so far, there has been a significant lack of research on the structural effects of alkali metal ions on nucleoside free radicals. In this study, we report a new method for generating metalized nucleoside radical cations in the gas phase. The radical cations [Ade+M-H]^•+^ (M = Li, Na) are generated by the 280 nm ultraviolet photodissociation (UVPD) of the precursor ions of lithiated and sodiated ions of 2-iodoadenine in a Fourier transform ion cyclotron resonance (FT ICR) cell. Further infrared multiphoton dissociation (IRMPD) spectra of both radical cations were recorded in the region of 2750–3750 cm^−1^. By combining these results with theoretical calculations, the most stable isomers of both radicals can be identified, which share the common characteristics of triple coordination patterns of the metal ions. For both radical species, the lowest-energy isomers undergo hydrogen transfer. Although the sugar ring in the most stable isomer of [Ade+Li-H]^•+^ is in a (South, syn) conformation similar to that of [Ado+Na]^+^, [Ade+Na-H]^•+^ is distinguished by the unexpected opening of the sugar ring. Their theoretical spectra are in good agreement with experimental spectra. However, due to the flexibility of the structures and the complexity of their potential energy surfaces, the hydrogen transfer pathways still need to be further studied. Considering that the free radicals formed directly after C-I cleavage have some similar spectral characteristics, the existence of these corresponding isomers cannot be ruled out. The findings imply that the structures of nucleoside radicals may be significantly influenced by the attached alkali metal ions. More detailed experiments and theoretical calculations are still crucial.

## 1. Introduction

DNA damage is one of the common problems faced by many organisms, especially when caused by radiation and oxidation. In these damage processes, base residues are often the main targets of high-energy particles, leading to double-strand breakage (DSB) and base oxidation, etc. Recent studies have shown that nucleoside radicals play a crucial intermediate role in RNA/DNA damage [1,2,3]. However, the high reactivity, extremely short lifespan, and instability of nucleoside radicals make it difficult to study their structures. To solve the problem, some methods for generating nucleoside radicals in the gas phase have been developed. Recently, spectroscopy studies combined with theoretical calculations have been used to observe nucleoside radical cations [4,5,6,7,8,9,10,11,12,13,14,15,16,17]. But the generation of nucleoside cation radicals frequently requires more demanding conditions, such as dissociative oxidation from ternary transition metal complexes [M(L3) (N)]^2+^(M: Cu (II), Pt (II); L: dien, tpy, Nu: A, dA, G, dG, C, dC) [4,5,6,7,8], charge-tagged chemical derivatization [9,10,11,12] and so on [13,14,15,16,17].

Alkali metal ions are a common group of ions in living organisms that participate in various biochemical reactions and physiological functions. These ions coordinate with the negatively charged part of the nucleoside radical, stabilizing the structure of the radical through charge interactions. Therefore, the study of the interaction between metal ions and nucleoside molecules has always received attention. In 1962, the ways in which metal ions affect the stability of double-stranded DNA was studied for the first time [18]. Subsequently, scientists have begun to conduct in-depth research on the interactions between metal ions and neutral [19,20,21,22,23,24], single [25,26,27] or double deprotonated nucleotides [28,29,30]. In this regard, Wang et al. [31] observed sodium cationization on four doubly deprotonated mononucleotide dianions, Na^+^·[dNMP-2H]^2−^ (N = A, G, C, or T), in the gas phase and showed that Na^+^ enables novel double deprotonation patterns. In addition, PavelHobza et al. [32,33] also reported three types of interactions between (N1 and N10), (N7 and N10) and (N3 and N9) of tautomers and cations, wherein the strongest interactions were observed for bifurcated coordination when the cation was located between the N3 and N9 atoms. However, how Li^+^/Na^+^ affect the structure of nucleobases [34] and nucleosides [35], especially nucleoside radicals, has not been studied in detail.

Here, by combing a Fourier transform ion cyclotron resonance (FT ICR) mass spectrometer with tunable infrared and ultraviolet lasers [36,37], the problem of the stable generation of adenosine radical cations is solved. By utilizing tunable infrared and ultraviolet lasers, we can excite 2-iodoadenine ions at specific wavelengths to obtain their infrared multiphoton dissociation (IRMPD) spectra. We can infer the structure and bonding characteristics of adenosine radical cations from their IRMPD spectra. In addition, in order to further verify the experimental results, the density functional theory (DFT) calculation method was used to search for the structures of adenosine radical cations.

## 2. Results

### 2.1. Generation of the Radical Cations

The lithiated and sodiated ions of 2-iodoadenine were applied here as the precursor ions for the generation of the radical cations of [Ade+M-H]^•+^ (M = Li, Na). The process is shown in the upper half of Figure 1. Briefly, the precursor ions of lithiated and sodiated 2-iodoadenine ([^I^Ade+M]^+^, M = Li, Na) were generated using the ESI method, separately. These ions were injected into the ICR cell through a quadrupole ion guide. Then, the stored waveform inverse Fourier transform (SWIFT) excitation pulse was applied to select the precursor ions. After the isolation of the precursor ions in the cell of the FT ICR mass spectrometer, the radical cations [Ade+Li-H]^•+^ (*m*/*z* 273.10) and [Ade+Na-H]^•+^ (*m*/*z* 289.08) were generated by the irradiation of a UV laser at 280 nm (Appendix A). The target radical ions were then further selected (Figure 2a,e) and characterized by collision-induced dissociation (CID)/IRMPD/ultraviolet photodissociation (UVPD)-MS^3^ (Figure 2b–d,f–h).

The CID and IRMPD mass spectra of the lithiated radical cation were close to each other (Figure 2b,c). Fragments due to the loss of one OH group (*m*/*z* 256.10) and the loss of one C_4_H_7_O_3_ unit through the cleavage of the sugar ring (*m*/*z* 170.07) were observed in both mass spectra. In addition, fragment ions at *m*/*z* 142.07 were also observed, due to the glycosidic bond cleavage and accompanying hydrogen transfer. For [Ade+Na-H]^•+^, the results were similar to those of [Ade+Li-H]^•+^, but there were still some differences. The CID mass spectrum of [Ade+Na-H]^•+^ showed the only fragment ion at *m*/*z* 186.04 due to the loss of the C_4_H_7_O_3_ unit. For its IRMPD spectrum, the fragment ion at *m*/*z* 272.08 due to the OH group loss was also observed. However, no fragment ion due to the glycosidic bond cleavage was observed in either the CID or IRMPD experiments. For both radical ions, a 5 s UV irradiation at 280 nm resulted in no fragmentation (Figure 2d,h).

### 2.2. Photodissociation Spectroscopic Study of Metalized Adenosine Radical Cations

The method of MS^3^ was applied here to obtain the IRMPD spectra of the radical cations. As shown in Figure 1, the target radical cations were generated by the UVPD of the precursor ions of [^I^Ade+M]^+^ (M = Li, Na) at 280 nm. After that, they were mass-selected and exposed to a tunable IR laser to obtain the IRMPD action spectra. Based on their fragmentation yields, the IRMPD action spectra of [Ade+M-H]^•+^ (M = Li, Na) were calculated according to *Int* = −ln *(I_parent_*/(∑*I_fragment_* + *I_parent_*)), where *I_parent_* is the intensity of the parent ion ([Ade+M-H]^•+^) and *I_fragment_* is that of the fragment ion, respectively.

Herein, by collecting the IRMPD mass spectra obtained under different IR wavelengths, the IRMPD spectra of [Ade+M-H]^•+^ (M = Li, Na) were recorded in the range of 2750–3750 cm^−1^ and shown in Figure 3. The IRMPD spectra of the two radical ions showed quite similar characteristics. Both spectra showed absorptions that peaked at ~3460, ~3580 and ~3710 cm^−1^. The dissociation yields at the three peaks were 89%, 45%, 79%, and 63%, 25%, 48% for [Ade+Li-H]^•+^ and [Ade+Na-H]^•+^, respectively. The 3710 cm^−1^ peaks indicate that free -OH groups exist in both ions. But there were also some differences. For [Ade+Na-H]^•+^, the strongest peak was located at 3468 cm^−1^, showing a 8 cm^−1^ blue shift compared with that of [Ade+Li-H]^•+^. A weak absorption peak at 3635 cm^−1^ was only observed in the case of [Ade+Li-H]^•+^. The spectra could be compared with the IRMPD spectra of [Ado+Na]^+^ and [dAdo+Na]^+^ previously reported by Rodgers et al. [35]. The spectra of [Ade+Li-H]^•+^ and [Ade+Na-H]^•+^ were similar to those of [Ado+Na]^+^ and [dAdo+Na]^+^, respectively. And in the latter cases, the weak absorption at ∼3600 cm^−1^ that could be applied to distinguish the two species in the region of 3300–3800 cm^−1^ was only observed for [Ado+Na]^+^ [35].

### 2.3. Structures of [Ade+Li-H]•^+^ and [Ade+Na-H]•^+^

To understand the structures of the radical cations, extensive DFT calculations were performed here. Several facts should be considered. First, isomers formed through different hydrogen transfer reactions after the homocleavage of the C-I bond should be considered systematically. Second, the coordination styles of the metal ions have great effect on energies. Based on previous reports [32,33], all the most important locations for the alkali metal cations to interact with the molecules, including bidentate coordination (N1 and N10, N7 and N10, N3 and O′) and tridentate coordination (N3, O′ and O5), are considered here (Appendix A; for the atomic number, please refer to Figure 1). Third, other structural factors should be considered, including the conformation of the sugar ring (north or south), the nucleobase orientation (anti or syn) and different intra-cluster H-bonds.

Based on the calculation strategy described above, the most stable isomers of [Ade+Li-H]^•+^ in different structural styles were identified at last and are shown in Figure 4. According to the above scenarios we considered, the isomer Li-N3-O′-O5′-C2H-1 was found to be the most stable structure. It is characterized by the occurrence of the hydrogen transfer reaction after the homocleavage of the C-I bond and its coordination style. Its counterpart isomer without hydrogen transfer (Li-N3-O′-O5′-1 in Figure 4) had a higher energy of 3.88 kcal/mol. Overall, our calculations show that the radical cations generated in situ (C2^•^) typically have relatively high energies. In most cases, the energy of the structure that undergoes hydrogen transfer after ultraviolet irradiation (Figure 4, top half) is 3–8 kcal/mol lower than that of the corresponding structure without hydrogen transfer (Figure 4, bottom half).

Since it is believed that a hydrogen transfer following the cleavage of the C-I bond could happen in the experiments [36], different isomers with different hydrogen atom donors have also been considered. Results show that the two lowest energy structures (Li-N3-O′-O5′-C2H-1 and Li-N3-O′-O5′-C2H-2 in Figure 4) had similar energies but were slightly different in their structures. The isomer of Li-N3-O′-O5′-C2H-1 can be thought to be formed through the hydrogen transfer from the hydrogen in the C2′ hydroxyl group to the C2 position after the homocleavage of the C-I bond by UV irradiation, while the isomer of Li-N3-O′-O5′-C2H-2 is formed through the hydrogen transfer from the C3’ hydroxyl group.

From the coordination point of view, the lowest energy structures (Li-N3-O′-O5′-C2H-1, -2, -3 in Figure 4) were all characterized by the tricoordinated metal ions that bind to two oxygen atoms (one in the sugar ring, one in the C5′ hydroxyl) and a nitrogen atom (the amino N3). Here, the isomer Li-N3-O′-O5′-C2H-3, which was 1.01 kcal/mol higher in energy than the lowest energy structure, was only different from the latter in the orientation of the sugar ring. The corresponding dihedral angles (C4-N9-C1′-C2′) in isomers of −3 and −1 were −149.3° and −64.4°, respectively. For complexes of Li-N3-O′, Li^+^ binds to the oxygen atom in C2′ hydroxyl and the amino N3. The most stable isomer of this family, Li-N3-O′-C2H-1, had an energy of 2.92 kcal/mol higher than the isomer of Li-N3-O′-O5′-C2H-1. Although the relative energy is not so high, it should be noted that this isomer is characterized by an accompanying break of the sugar ring. For isomers with other coordination styles (Li-N7-N10- and Li-N1-N10- in Figure 4), their relative energies were much higher (>22 kcal/mol).

Considering the conformation effect, it was found that the most stable structure (Li-N3-O′-O5′-C2H-1) had a (south, syn) conformation. Its north- counterparts have not been identified here. Similarly, the lowest energy isomers with two coordination sites (Li-N3-O′-O5′-C2H and Li-N7-N10-C2H in Figure 4) all had a (south, syn) conformation, although the isomers of Li-N3-O′-C2H and Li-N1-N10-C2H had a south-anti conformation.

Figure 5 compares the measured IR spectrum with the calculated spectra of different isomers obtained at the level of B3LYP/6-311+G(d,p). For the convenience of comparison, the predicted IR spectra of the isomers with or without hydrogen transfer, corresponding to the upper and lower parts of Figure 4, are displayed on the left and right sides of Figure 5, respectively. For the three lowest-energy structures, Li-N3-O′-O5′-C2H-1, -2 and -3, the stretch modes of the free OH groups all matched the experimental band position of 3710 cm^−1^ very well. Both NH_2_ symmetric stretching vibrations appearing at ~3450 cm^−1^ and asymmetric stretching vibrations appearing at ~3580 cm^−1^ were found to be in good agreement with the measured bands centered at 3460 and 3580 cm^−1^, respectively. For Li-N3-O′-O5′-C2H-1 and -3, they both had one H-bonded OH group, which had absorption at ~3635 cm^−1^, agreeing very well with the experimental weak peak at 3635 cm^−1^. The isomer Li-N3-O′-O5′-C2H-2 also had an H-bonded OH group, but its peak was blue-shifted more and merged in the peak at 3580 cm^−1^. Thus, the comparison among the experimental and calculated spectra clearly reflected that although all three top structures may contribute to the experimental spectrum, the isomer Li-N3-O′-O5′-C2H-2 accounted for the majority. The assignment of the observed peaks is summarized in Appendix A. For the isomer of Li-N3-O′-C2H-1, although its theoretical spectrum matched the experimental spectrum very well, its free energy was 2.2 kcal/mol higher than the most stable one, and its contribution should be insignificant. The top isomers without hydrogen transfer (Li-N3-O′-O5-1 and Li-N3-O′-O5-2) had relative energies greater than 3.88 kcal/mol, but their contributions cannot be ruled out here. For one thing, although hydrogen transfer following the UV cleavage of C-I bonds has been observed in some previous examples [36], the homolysis does not always lead to isomerization [38,39]. It should be mentioned that the anharmonicity of the H-bonded OH stretch is not considered here. Since the harmonic approximation simulations might bring some mistakes in some cases [40,41], that factor should be considered here. Possible isomerization pathways were tried here, but due to the flexibility of the species and the complexity of its potential energy surface, the pathway with the lowest energy barrier is still being sought. Other isomers with different coordination styles can be neglected, since not only are their energies much higher, but their spectra also do not match the experiment result.

For [Ade+Na-H]^•+^, we also considered its stable isomers/conformations in different styles in a manner similar to that of [Ade+Li-H]^•+^. The isomer Na-N3-O′-O5′-C2H-1 was obtained via the optimization of the initial structure constructed by replacing the Li atom with a Na atom in the most stable isomer of [Ade+Li-H]^•+^. It had a very similar structure to that of Li-N3-O′-O5′-C2H-1. Due to differences in the size of the metal ions, the lengths of the three coordination bonds corresponding to Na^+^ were all longer than those corresponding to Li^+^. Meanwhile, the dihedral angle of C4-N9-C1′-C2′ was –53.5°, smaller than that in the case of Li^+^ (−64.4°).

For convenience of discussion, the energies of other isomers of [Ade+Na-H]^•+^ were all discussed relative to that of the isomer Na-N3-O′-O5′-C2H-1, and these values are shown in Figure 6. Similarly, the energies of the structures that undergo hydrogen transfer (Figure 6, above half) were at least 3.80 kcal/mol lower than those of the corresponding structures without hydrogen transfer (Figure 6, below half). For coordinated isomers at three positions of N3, O′ and O5′, the isomer Na-N3-O′-O5′-C2H-2 was 0.06 kcal/mol lower in energy than Na-N3-O′-O5′-C2H-1. Considering its dihedral angle of C4-N9-C1′-C2′ of −161.1°, it can be concluded that the size of metal ions and the dihedral angle could have an impact on its energy together. For isomers without hydrogen transfer, a similar relationship has been observed. Most importantly, it has been found that the structure of Na-N3-O′-C2H-1, which is characterized by an unexpected cleavage of the sugar ring, is the most stable isomer. It has an energy (Gibbs free energy) of 5.25 (6.18) kcal/mol lower than that of Na-N3-O′-O5′-C2H-1. And other isomers have been found to be much more unstable than it.

Figure 7 compares the measured IR spectrum with the calculated spectra of these isomers shown in Figure 6. The calculation was performed on the level of B3LYP/6-311+G(d,p). For the lowest-energy structure, Na-N3-O′-1-C2H, it could be found that the predicted IR spectrum matched the experimental one best. In addition to the stretch modes of free OH at ~ 3710 cm^−1^, the predicted NH_2_ symmetric and asymmetric vibration modes at 3457 and 3575 cm^−1^ both matched the experimental peaks at 3468 and 3580 cm^−1^ well. It should be noted that the experimental peak at 3468 cm^−1^ was blue-shifted by 8 cm^−1^ compared to the peak at 3460 cm^−1^ in the case of [Ade+Li-H]^•+^, while its theoretical peak at 3457 cm^−1^ was blue-shifted by 7 cm^−1^ compared to the peak at 3450 cm^−1^ in the latter case. Another point is that the absence of the weak peak at 3635 cm^−1^ compared to the spectrum of the lithiated radical also indicated the difference in the sodiated radicals. Nevertheless, considering other facts, such as the influence of anharmonicity, poor intramolecular vibrational redistribution (IVR), and the exited isomerization barriers, the contributions from other isomers, such as Na-N3-O′-O5′-C2H-1, -2 or Na-N3-O′-O5′-1, -2, should not be neglected here. For the predicted peaks in the region of 2850–3200 cm^−1^, no experimental peak was observed for either radical ion. Typically, these calculated peaks originate from the vibration of CH bonds. Considering several possible reasons, such as the low activity of the bands, poor IVR and the coexisting multiple isomers, these peaks were hard to detect in the experiment.

## 3. Discussion

According to the analysis shown above, some conclusions can be drawn here. First, for both [Ade+Li-H]^•+^ and [Ade+Na-H]^•+^, their most stable isomers tended to undergo hydrogen transfer after the homocleavage of the C-I bond due to ultraviolet irradiation. Second, the most stable structures in both cases were characterized by three-coordination formed by the combination of M^+^ with two oxygen atoms (one in the sugar ring, one in the C5 ′ hydroxyl group) and one nitrogen atom (amino N3). It can be seen that M^+^ cations can promote the folding of the whole species by combining with nucleobases and sugar rings via chelation. The coordination between adenosine radicals and Li^+^/Na^+^ is much more complex than that between bases and alkali metal cations, which has been reported by other research groups. For such complexes, it has been reported that there are three types of interactions between (N1 and N10), (N7 and N10) and (N3 and N9) of tautomers and cations, wherein the last type has the strongest interaction [32,33].

In addition to considering the three coordination situations mentioned in the literature, it is also necessary to consider whether hydrogen transfer occurs after the homocleavage of the C-I bond, the conformation of the sugar ring (North or South), the nucleobase orientation (anti or syn) and different intra-cluster H-bonds. Third, it was found that the most stable structure (Li-N3-O′-O5′-C2H-1) has a (south, syn) conformation. This can be compared with previous studies on relevant species. For example, Wang et al. [31] showed that Na^+^ combines with deprotonations from the phosphate and the hydroxyl group in sugar to form a tetradentate north-syn conformation. Rodgers et al. [35] found that the [dAdo+Na]^+^ and [Ado+Na]^+^ complexes are formed by sodium cations binding to the N3, O′ and O5′ atoms. They had a (south, syn) conformation, which is similar to the most stable structure of [Ade+Li-H]^•+^ reported here.

Last but not least, the theoretical IRMPD action spectrum of [Ade+Na-H]^•+^ shows that the adenosine radical cation observed in the experiment is Na-N3-O′-C2H-1. It is worth mentioning that its characteristic is the opening of the sugar ring, which can be supported well in the CID/IRMPD-MS^3^ mass spectra. As shown in Figure 2, the fragment ion due to the glycosidic bond cleavage, which has been observed in the case of [Ade+Li-H]^•+^ (*m*/*z* 142), was totally absent in the CID/IRMPD experiments of [Ade+Na-H]^•+^, indicating that a stronger glycosidic bond existed in the sodiated radical cations. This point is further proved by the calculated structures shown in Figure 4 and Figure 6. As shown there, it was found that the lengths of the glycosidic bonds in the most stable isomers of [Ade+Li-H]^•+^ and [Ade+Na-H]^•+^ (Li-N3-O′-O5′-C2H-1, -2 -3 and Na-N3-O′-C2H-1) were 1.42–1.46 and 1.39 Å, respectively. To further verify the conclusion that the sodium ion can induce the cleavage of the sugar ring while the lithium ion cannot, we also conducted corresponding calculations on K^+^. The results show that the isomer K-N3-O′-O5′-C2H-1, which is also characterized by the opening of the sugar ring, had an energy (free energy) 5.29 (5.86) Kcal/mol higher than that of the isomer K-N3-O′-C2H-1 (Appendix A, showing that K^+^ has a similar structure effect to that of Na^+^.

## 4. Materials and Methods

### 4.1. Mass Spectrometry and Photodissociation

All experiments were carried out using a self-designed experimental setup combining a double-beam laser system and an FT ICR mass spectrometer. The details of the experimental setup have been described previously [37]. The mass spectra weremeasured using the 7.0 T FT ICR mass spectrometer (Varian IonSpec, Lake Forest, CA, USA) equipped with a commercial Z-spray ion source. The 2-iodoadenine−Li^+^/Na^+^ complex ions, [^I^Ade+M]^+^ (M = Li, Na), were prepared from solutions of 1.0 mM 2-iodoadenine in H_2_O/MeOH/LiCl (or NaCl) (49:49:2; *v*/*v*/*v*) with a flow rate of 2 μL/min through the Z-Spray ESI source. The sample of 2-iodoadenine was purchased from Aladdin (Shanghai, China) and used without further purification.

The generated ions were injected into the ICR cell via a quadrupole ion guide. In the ICR cell, the target ions were mass-selected using the method of stored waveform inverse Fourier transform (SWIFT) [42]. N_2_ was used as the cone and desolvation gases. The voltages of the probe and cone were set to be 3.8 kV and 22 V, respectively. For CID experiments, the method of sustained off-resonance irradiation (SORI) excitation of the selected ions was applied [43,44], with a typical amplitude of 2.2 V (V_p−p_) and a frequency offset of 1000 Hz.

For the double-beam laser system, a UV OPO laser (NT-342C, EKSPLA, Vilnius, Lithuania) and an IR OPO laser (Firefly-IR, M Squared, Glasgow, UK) were applied. In the experiments reported here, the UV laser was set at 280 nm, with a frequency of 10 Hz and an output energy of 2 mJ/pulse. The IR OPO laser was operated with an output irradiation tunable laser from 2750 to 3750 cm^−1^ with a line width of 7 cm^−1^ and a typical output energy of 100 mW. The adjustable triggers were produced by a pulse generator (DG535, Stanford, CA, USA), the outputs of which were linked to a multichannel mechanical shutter controller (SSH-C4RA, Sigma-Koki, Tokyo, Japan) to control the irradiation times of the two lasers.

### 4.2. Computational Details

All computations were performed with the Gaussian 09 program [45]. Geometries for complexes were initially optimized at the B3LYP/6-31+G(d) level. Vibrational frequencies were calculated from these optimized structures. Additionally, geometry optimizations and vibrational frequencies of the lowest energy structures of each coordination site were further performed at the level of B3LYP-D3/6-311+G(d,p). A single scaling factor of 0.964 was applied to the calculated harmonic frequencies in the 2750–3750 cm^−1^ region.

## 5. Conclusions

In summary, a new method for the generation of metalized nucleoside radical cations has been developed and applied here. Via the UVPD of the precursor ions of [^I^Ade+M]^+^ (M = Li, Na) at 280 nm, the radical cations of [Ade+M-H]^•+^ (M = Li, Na) were steadily generated. IRMPD spectra of [Ade+M-H]^•+^ (M = Li, Na) were then recorded in the region of 2750–3750 cm^−1^. The experimental spectra of both radical cations showed that absorptions peaked at ∼3460 cm^−1^, ∼3580 and ∼3710 cm^−1^. For [Ade+Li-H]^•+^, a weak absorption peak at 3635 cm^−1^ was also observed. Theoretical calculations were systematically carried out to identify the most stable isomers of the corresponding radical cations. For both radicals, the predicted IR spectra of the most stable isomers agreed with the experimental spectra. The most stable isomers of both species were characterized by two factors: one is the hydrogen transfer after the homo-cleavage of the C-I bond; the other is that the metal ions in both radicals exhibited similar triple coordination patterns, which promote the folding of mononucleosides via multipronged chelation to the nucleobase and sugar ring. The two species were also characterized by their structural differences. Among them, the most stable isomer of [Ade+Li-H]^•+^ had a (south, syn) conformation of the sugar ring, which was similar to that of [Ado+Na]^+^ previously reported by Rodgers et al. [35]. Interestingly, the isomer of [Ade+Na-H]^•+^ was characterized by the opening of the sugar ring, indicating that the sizes of the alkali metal ions can have great effects on the structures of the radicals. Although their energies are 2–4 kcal/mol higher than the most stable ones, the C2^•^ radials generated by the direct homo-cleavage of the C-I bond cannot be ruled out in these studies. Their corresponding isomerization pathways for the C2^•^ radials to form the most stable isomers still need further investigation.

## Figures and Tables

**Figure 1 ijms-24-15385-f001:**
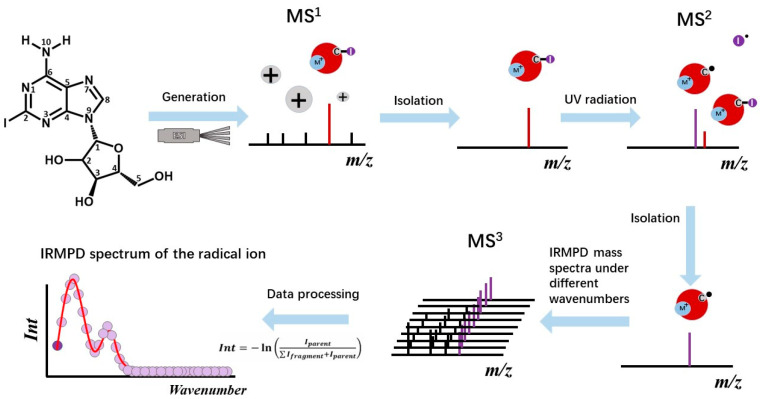
Experimental flowchart showing how to obtain the IRMPD spectra of radical cations of [Ade+M-H]^•+^ (M = Li, Na).

**Figure 2 ijms-24-15385-f002:**
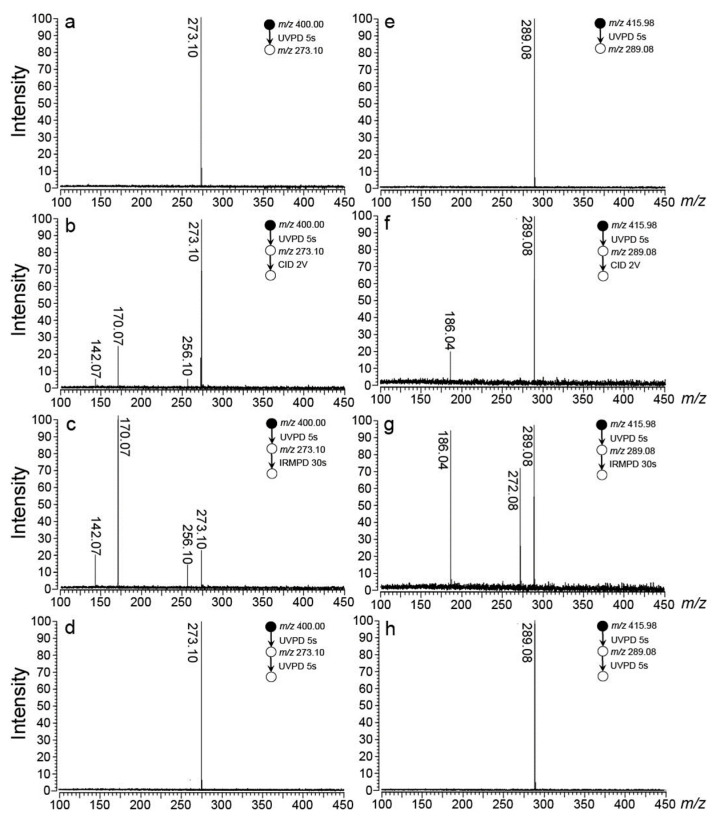
Mass spectra and tandem mass spectra of [Ade+M-H]^•+^ (M = Li, Na). Among them, (**a**–**d**) are for [Ade+Li-H]^•+^ and (**e**–**h**) are for [Ade+Na-H]^•+^. They are the mass spectra of radical cations after (**a**,**e**) isolation; (**b**,**f**) tandem CID experiments; (**c**,**g**) tandem IRMPD experiments; and (**d**,**h**) tandem UVPD experiments. The wavelength of the UV laser applied here is 280 nm, and the wavenumber of the IR laser applied here is 3468 cm^−1^. The irradiation periods for UV and IR lasers are 5 and 30 s, respectively.

**Figure 3 ijms-24-15385-f003:**
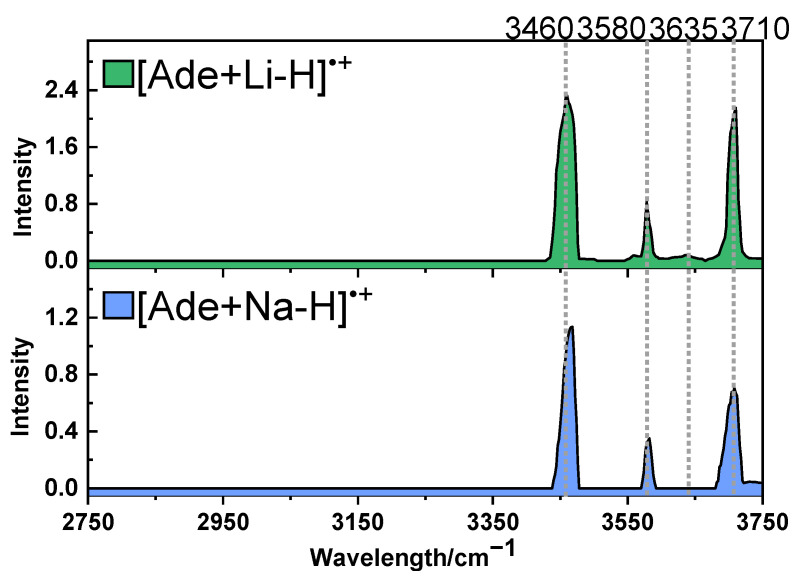
Experimental IRMPD spectra of [Ade+Li-H]^•+^ (**above**) and [Ade+Na-H]^•+^ (**below**).

**Figure 4 ijms-24-15385-f004:**
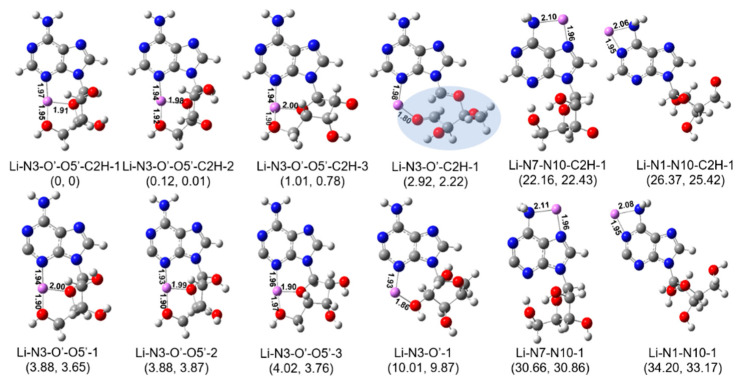
Minimum energy structures of [Ade+Li-H]^•+^, calculated at the B3LYP-D3/6-311+G(d,p) level. Isomers with different coordination sites are considered. The isomers shown in the above half are all characterized by hydrogen transfer, while those in the below half are not. Their relative energies and Gibbs free energies (in kcal/mol) are shown in parentheses. The lengths of some bonds (in Å) are shown in corresponding structures for convenience of comparison. The shadowed region indicates a cleavage of the sugar ring. Figure 1 can be referred to for the atom labels used in the names of the isomers.

**Figure 5 ijms-24-15385-f005:**
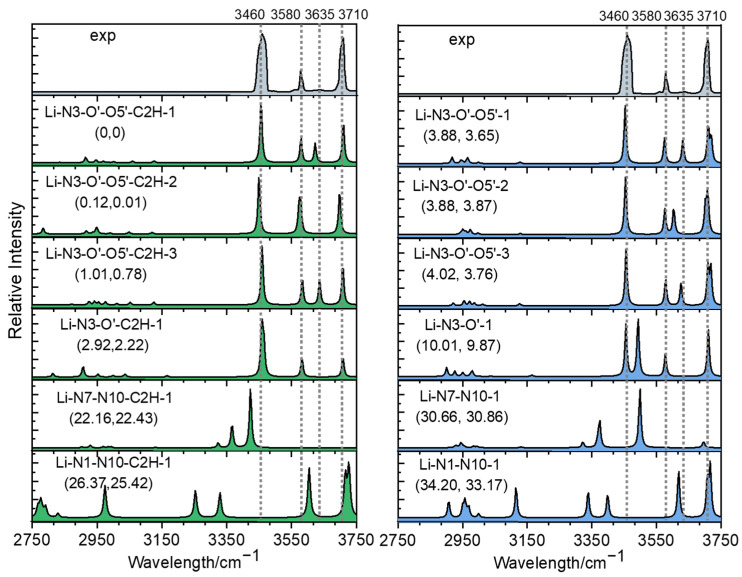
Comparison among the experimental IRMPD spectrum and calculated absorption spectra of [Ade+Li-H]^•+^ in the region of 2750−3750 cm^−1^. The calculation was completed on the level of B3LYP-D3/6-311+G(d,p). The left and right columns correspond to the isomers shown in the above and below halves of Figure 4, respectively. The relative energies and Gibbs free energies (in kcal/mol) of these isomers are shown in parentheses in the upper left corner of each subgraph.

**Figure 6 ijms-24-15385-f006:**
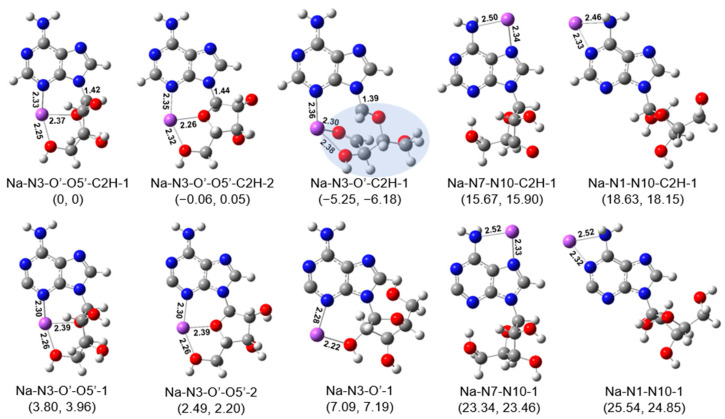
Minimum energy structures of [Ade+Na-H]^•+^, calculated at the B3LYP-D3/6-311+G(d,p) level. Isomers with different coordination sites are considered. The isomers shown in the top half are all characterized by hydrogen transfer, while those in the bottom half are not. Their relative energies and Gibbs free energies (in kcal/mol) are shown in parentheses. The lengths of some bonds (in Å) are shown in corresponding structures for convenience of comparison. The shadowed region indicates a cleavage of the sugar ring. The atom labels used in their names can be found by referring to Figure 1.

**Figure 7 ijms-24-15385-f007:**
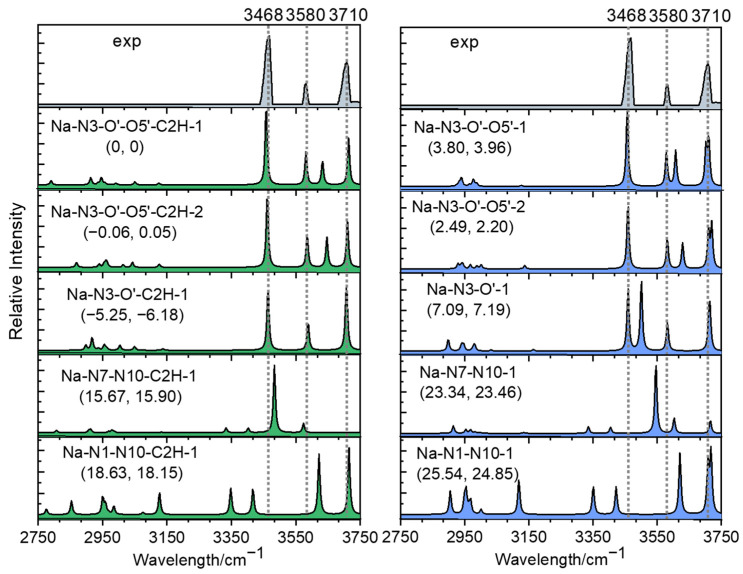
Comparison between experimental IRMPD spectrum and calculated absorption spectra of [Ade+Na-H]^•+^ in the region of 2750−3750 cm^−1^. The calculation was completed on the level of B3LYP-D3/6-311+G(d,p). The left and right columns corresponded to the isomers shown in the above and below halves of Figure 6, respectively. The relative energies and Gibbs free energies (in kcal/mol) of these isomers are shown in parentheses in the upper left corner of each subgraph.

## Data Availability

The data are available on request from the corresponding author.

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
