# Peer review of "Structural Characterization of the Metalized Radical Cations of Adenosine ([Ade+Li-H]•+ and [Ade+Na-H]•+) by Infrared Multiphoton Dissociation Spectroscopy and Theoretical Studies"

_ijms, 2023, doi:10.3390/ijms242015385_

Round 1

Reviewer 1 Report

The manuscript is well written and the study is well described. The conclusions are certainly valid and based on the results.

How does the work compare to other IRMPD studies of complexes between adenine and alkali metal cations that are not referenced?

2-I-adenine is not defined anywhere? The proper chemical name would be 2-iodoadenine.  2-I-adenine is confusing and should not be used.

Reviewer 2 Report

The authors have explored the structure of sodiated and lithiated complexes of radical adenosine produced by UVPD of the corresponding complexes with 2-I-adenosine by MS and IRMPD. The results could be of some interest for the spectroscopy community, but less for the general audience of IJMS. However, I will support publication of the manuscript after major revisions.

Altough the comparison of the experiments with calculated IR spectra could support the structural attribution of the ions, I am not fully convinced. I think the author should responde to the following points:

- I don't think that the authors can safely assert the absence of Li-N3-'-O5'-1 from the gas-phase population of the [Ade+Li-H].+ ion on the basis of its relative energy alone. Its calculated spectrum  is that better interpreting the experimental spectrum of [Ade+Li-H].+. and the H-transfer process may be kinetically hindered. Please provide references for the sentence "Since it is believed that a hydrogen transfer following the cleavage of the C-I bond can  happen in the experiments, different isomers with different hydrogen atom donors have also been considered." at line 159. In addition, calculations of the energy needed for the H-transfer should be performed.

- The formation of Na-N3-O'-C2H-1 after photoactivation of [2I-Ade+Na]+ should be supported by calculations of the energy involved in the process. In addition, the authors must report the dissociation yield of the IRMPD spectrum. In fact, the absence of any band at 3600 cm-1 could derive from several factors which are related to the IRMPD process (poor IVR, low activity of the band, anharmonicity, etc.). Therefore generating an experimental bias in the attribution of the bands. 

Additional minor remarks:

- Considering the energy in place, the IR dissociation process is likely multiphotonic. Please replace in the whole text and in figure 1 IRPD with IRMPD.

- The author should consider to add in the SI tables with the vibrational mode assignment.

English is overall okay. Some suggestions for improvement are below:

- Line 112: the first sentence needs rephrasing.

- Line 115: please replace extents with yields

- line 174: " the most stable isomer of this family"

- line 291: replace "si" with "is"

- Reference 11: correct "Turecek"

Reviewer 3 Report

The paper by Kou et al. describes a study of adenosine radicals complexed with Li+ and Na+ cations, which are generated by UV photodissociation of iodinated precursors and investigated using infrared multiphoton dissociation spectroscopy experiments and density functional theory calculations. The authors report evidence that the radical cations are formed, and then undergo H atom transfer. One of the clusters may have a cleaved sugar ring. This paper investigates biologically important radical ions, which have very flexible structures that make analysis complicated. The authors do a good job of exploring this structural complexity, and present nice-looking spectra. However, there are some points that I think should be addressed before this article should be published.

1)    I think that it is important authors mention work on the formation of radicals and radical cations from C-I bond homolysis, which does not lead to isomerisation (for example, from Kaiser and co-workers and from Trevitt and co-workers). I believe that the flexible molecules in this study are more likely to undergo H atom transfer. Nevertheless, the fact that C-I bond cleavage very often selectively creates radical sites without isomerisation is important.

2)    The agreement between the experimental spectra and the simulations is good for the NH2 stretches at 3460 cm-1 and 3580 cm-1 as well as the OH stretch at 3710 cm-1 for several isomers. The key feature for comparison between experiment and theory is at 3635 cm-1, which is either absent or weak in the experiment. The authors mention that the bands near 3635 cm-1 in the calculated spectra arise from H bonded OH stretches. If I understand correctly, these H bonded OH stretches should be quite anharmonic, which can make the simulations within the harmonic approximation unreliable. Two papers that feature extreme examples of similar effects are: Chen, L., Ma, Z. and Fournier, J.A., 2022. Origins of the diffuse shared proton vibrational signatures in proton-coupled electron transfer model dyad complexes. The Journal of Chemical Physics157(15). and Bell, M.R., Cruzeiro, V.W.D., Cismesia, A.P., Tesler, L.F., Roitberg, A.E. and Polfer, N.C., 2018. Probing the structures of solvent-complexed ions formed in electrospray ionization using cryogenic infrared photodissociation spectroscopy. The Journal of Physical Chemistry A122(37), pp.7427-7436. Therefore, it is possible that some of the harmonic approximation simulations might incorrectly feature a strong OH stretch at 3635 cm-1 as an artefact of the harmonic approximation. I understand that it is probably unreasonable to undertake simulations that consider anharmonic effects for this study because of the complexity of the molecule. Nevertheless, the conclusions of the paper rely considerably on the absence of the 3635 cm-1 band in the experiment, and the anharmonicity of the H bonded OH stretch is not mentioned. I think it is necessary to have some discussion about the anharmonicity of this mode. This is even more important for the lithiated ion, which has some signal at 3635 cm-1 in the experimental spectrum.

3)    In my opinion, it is not possible to rule out the Li-N3-O’-O5-1 or Li-N3-O’-O5-2 isomers of the lithiated adenosine radical. This is for three reasons: (1) as mentioned above, the band at 3635 cm-1 may be anharmonic enough that it appears erroneously intense and sharp in the simulation, while it may appear broader and weaker in the spectrum—furthermore, there is some signal in the experimental spectrum at 3635 cm-1, (2) because these isomers are not that much higher in energy than the lowest energy isomer—especially considering these are open shell species, which makes the calculations slightly less accurate, and (3) these are the isomers that are expected to be formed directly from homolysis of the C-I bond.

4)    The authors emphasise that the Na-N3-O’-C2H-1 isomer is characterised “by an unexpected cleavage of the sugar ring”. This cleavage is not evident in the structure of this isomer in Figure 6.

5)    There are several weak transitions in the calculated spectra between 2750 cm-1 and 3300 cm-1 for every isomer. However, there seems to be absolutely no signal in the experimental spectrum within this energy range. I think some discussion about is important.

6) perhaps it would be good to include Figure S2 from the supporting information into Figure 1?

The quality of english in this paper is quite good. I recommend some minor changes

1)    Line 291 on page 9 the word “si” should probably be “is”

2)    In my opinion, on line 127 of page 4, the words “Besides, it is worth noting that” should be omitted.

3)    Figures 5 and 7 should be introduced in the text before they appear

4)    Line 348 “in well agreement” should be “in agreement”

Round 2

Reviewer 2 Report

I thank the authors for their response which have addressed all of my comments.

I now support publication on IJMS after corrections of some flaws in the english of the added parts.

I think that the revised parts of the manuscript need some rephrasing.

From line 221 to line 230 the overall english needs to be adjusted. Simpler sentences, there is no need to use "although" and "however" every two lines. Lines 221 and 228 were copied and pasted... Please use a little bit of imagination...

The same apply to the conclusions.

Reviewer 3 Report

I am satisfied with the Authors' responses, I only have a few additional minor suggestions.

1) lines 221 to 223 "Thus, the top isomers without hydrogen transfer, although their energies are at least 3.67 kcal/mol higher, however, their contributions, such as Li-N3-O'-O5-1 or Li-N3-O'-O5-2, cannot be neglected or ruled out here." should be revised to improve its clarity. 

2) The sentence from lines 221 to 223 is repeated somewhat in lines 228 to 230. This sentence (228-230) should be revised to reduce repetitiveness and improve clarity. 

3) In my previous review I asked the Authors to include a reference from Trevitt and coworkers, and they included ref. [39] by Marlton and Trevitt. A more appropriate reference for C-I bond photodissociation with negligible isomerisation from Trevitt and coworkers is probably this one: Shiels, O.J., Kelly, P.D., Bright, C.C., Poad, B.L., Blanksby, S.J., Da Silva, G. and Trevitt, A.J., 2021. Reactivity trends in the gas-phase addition of acetylene to the N-protonated aryl radical cations of pyridine, aniline, and benzonitrile. Journal of the American Society for Mass Spectrometry32(2), pp.537-547.

As mentioned above, the authors should revise lines 221 to 223, and lines 221 to 223 because these lines are unclear, repetitive, and contain too many commas (in my opinion).
